# MCP-1 Reduction by L-SIGN Expression in Dengue Virus-Infected Liver Endothelial Cells

**DOI:** 10.3390/v17030344

**Published:** 2025-02-28

**Authors:** Keh-Sen Liu, Lin Wang, Po-Ming Chen, Ing-Kit Lee, Kuender D. Yang, Rong-Fu Chen

**Affiliations:** 1Division of Infectious Diseases, Department of Internal Medicine, Show Chwan Memorial Hospital, Changhua 500, Taiwan; milka2@msn.com; 2Department of Pediatrics, Pojen Hospital, Kaohsiung 807, Taiwan; 3Research Assistant Center, Show Chwan Memorial Hospital, Changhua 500, Taiwan; yaoming9@yahoo.com.tw; 4Department of Nursing, Central Taiwan University of Science and Technology, Taichung 406, Taiwan; 5Division of Infectious Diseases, Department of Internal Medicine, Chang Gung Memorial Hospital Kaohsiung Medical Center, Kaohsiung 803, Taiwan; 6Department of Medical Research, MacKay Memorial Hospital, Taipei 104, Taiwan; 7Department of Pediatrics, MacKay Memorial Hospital, Taipei 104, Taiwan; 8Department of Medicine, MacKay Medical College, New Taipei 252, Taiwan; 9Division of Plastic Surgery, Department of Surgery, Kaohsiung Medical University Hospital, Kaohsiung 807, Taiwan; 10Regenerative Medicine and Cell Therapy Research Center, Kaohsiung Medical University, Kaohsiung 807, Taiwan

**Keywords:** dengue virus, L-SIGN, MCP-1

## Abstract

(1) Background: The C-type lectin domain family 4 member M (CLEC4M, also known as L-SIGN) is a crucial pathogen-recognition receptor for the dengue virus (DENV). Our previous study has exhibited a polymorphism in its extracellular neck region, specifically within the long tandem repeats of exon 4, which correlates with DHF in DENV infection and causes liver damage. (2) Methods: Using monocyte-derived dendritic cells (MDDCs) and SK-HEP1 liver endothelial cell lines to compare viral replication relative to L-SIGN expression. (3) Results: Results indicated that SK-HEP1 cells were more susceptible to DENV infection than MDDCs, and L-SIGN transfection significantly increased viral replication in SK-HEP1 cell lines. The study also found that L-SIGN-enhanced DENV infection is mediated by the decrease in monocyte chemoattractant protein-1 (MCP-1) but not interferon gamma inducible protein-10 (IP-10). These findings reveal that L-SIGN-induced DENV infection leads to reduced MCP-1 levels, which, in turn, enhances DENV replication velocity. (4) Conclusions: This study offers insights into the molecular mechanisms of DENV replication and identifies potential therapeutic targets involving MCP-1 and L-SIGN pathways.

## 1. Introduction

Dengue is the most widespread virus transmitted by mosquitoes. It is estimated that more than 390 million dengue infections occur globally each year, with around 96 million of these resulting in illness [1]. Symptomatic dengue infections can manifest in a variety of clinical forms, ranging from mild febrile illness to severe and potentially life-threatening conditions. The spectrum of dengue’s clinical presentations includes mild dengue fever, dengue hemorrhagic fever (DHF), dengue shock syndrome (DSS), and liver dysfunction [2,3,4].

There are four closely related but serologically distinct dengue viruses (DENV), known as DENV-1, DENV-2, DENV-3, and DENV-4. These viruses belong to the genus Flavivirus. Each serotype is unique, and infection with one serotype typically provides lifelong immunity against that specific serotype but not against the others [5,6,7,8,9,10,11]. The manifestations of dengue disease in each infected individual are believed to be influenced by a combination of both viral and host factors, including serotype variation, viral load, host immune response, prior dengue infection, and genetic factors [12].

DENV infects human leukocytes through specific lectin-like receptors, initiating a cascade of immune responses that can lead to severe and hemorrhagic manifestations. Two primary receptors involved in this process are DC-SIGN (dendritic cell-specific ICAM-3 grabbing non-integrin, encoded by the CD209 gene) and mannose receptors [13]. Unlike DC-SIGN, which is primarily expressed on dendritic cells, L-SIGN (liver/lymph node-specific ICAM-3 grabbing non-integrin, encoded by the CLEC4M gene, also known as CD299 or DC-SIGNR) has a distinct expression pattern and is found in different tissues such as the liver, lymph nodes, and placenta [14,15]. DC-SIGN plays a pivotal role in mediating dengue virus infection and initiating a series of intracellular signaling events that result in cytokine secretion. Additionally, the receptor is involved in transducing an immune evasion strategy employed by the virus to persist and proliferate within the host [16,17,18].

Viruses produced in primary dendritic cells (DCs) exhibit an inability to interact with the DC-SIGN receptor, although they remain infectious for cells expressing the L-SIGN receptor. This distinction is significant in understanding the dynamics of DENV infection and replication within the human body [19]. Replication of the DENV genome has been demonstrated both in vivo and in vitro across a variety of cell types, including liver hepatocytes and several types of hematopoietic cells [20]. The attachment of the dengue virus to the cell surface, followed by its entry into the host cell, represents the initial and critical step in a complex cascade of interactions between the virus and the target cell [21]. This step is a crucial determinant of tissue tropism and pathogenesis, making it a key target for antiviral host cell responses, such as antibody-mediated virus neutralization and antiviral therapy.

Both DC-SIGN and L-SIGN share the capacity to bind high mannose oligosaccharides through their carbohydrate recognition domains (CRDs) and are known to recognize a wide array of microbes, including HIV-1, Ebola virus, Hepatitis C virus, severe acute respiratory syndrome-associated coronavirus (SARS-CoV), and *Mycobacterium tuberculosis*. Additionally, both receptors possess a neck region characterized by varying lengths due to polymorphisms, with 69 base pairs (23 amino acids) tandem repeats in exon 4 [22,23,24,25]. The neck region is crucial for assembling both lectins into a tetrameric protein conformation on the cell surface. The length of this neck region potentially influences the pathogen-binding properties of these lectin receptors, affecting their ability to interact with various pathogens [15].

Our recent study revealed a significant correlation between the presence of L-SIGN and an increased risk of DENV developing dengue hemorrhagic fever [26]. The findings suggest that individuals exhibiting L-SIGN are more susceptible to DHF, indicating a possible link between this marker and the severity of dengue infections [26]. To investigate this correlation, we employed an optimized in vitro dengue virus replication model using monocyte-derived dendritic cells (MDDCs) and human liver endothelial cells. We further explored whether DENV induces differential levels of interferon-gamma inducible protein-10 (IP-10) and monocyte chemoattractant protein-1 (MCP-1) based on the expression of L-SIGN in human liver endothelial cells, specifically the SK-HEP1 cell line. We conducted an examination of the mechanism by which DENV amplifies cytokine production by elevating L-SIGN levels in SK-HEP1 cells. Our study focused on understanding how increased expression of L-SIGN influences the cellular response to DENV infection, specifically regarding the upregulation of pro-inflammatory cytokines.

## 2. Materials and Methods

### 2.1. Generation of DCs from Individuals with AA or AG Phenotype of rs4804803 and Infection with DENV-2

Peripheral blood mononuclear cells were collected from the peripheral blood of 5 healthy, DENV-specific IgM or IgG seronegative volunteers. CD14^+^ monocytes were isolated by positive selection according to the manufacturer’s specifications, using CD14 microbeads and a magnetic cell separator (MACS) (Miltenyi Biotec, Bergisch Gladbach, Germany). Enriched CD14^+^ cells (purity > 95% as determined by FACS analysis using BD Biosciences and data processing with CellQuest software version 3.0 from Becton Dickinson, San Jose, CA, USA) were cultured for 6 days in six-well plates in RPMI 1640 medium (GIBCO, Grand Island, NY, USA) supplemented with 10% FBS, 10 units/mL penicillin, and 10 μg/mL streptomycin in the presence of 10 ng/mL recombinant human granulocyte macrophage-colony stimulating factor (rhGM-CSF) and 5 ng/mL recombinant human interleukin-4 (rhIL-4) at 37 °C, 5% CO_2_, and 95% relative humidity. On day 3, half of the medium was replaced with fresh medium supplemented with rhGM-CSF and rhIL-4. Non-adherent cells were harvested. Expression of markers was measured by FACS using specific antibodies and their corresponding isotype controls.

Unless otherwise stated, MDDCs were infected with DENV-2 at a multiplicity of infection (MOI) of 5 for 2 h at 37 °C and 5% CO_2_. Cells were washed twice to remove cell-free virus, and cultured in complete RPMI medium (without cytokines) at a density of 2 × 10^5^ cells/mL in 48-well plates. Cells and supernatant were removed and analyzed at 24, 48, and 72 h post-infection.

### 2.2. Construction of L-SIGN Expression Plasmid and Transfection into Cell Lines

The wild-type L-SIGN cDNA, which contains seven tandem repeats in the neck region and is cloned into the pcDNA3.1 vector (Invitrogen Inc., Carlsbad, CA, USA), was derived from the original plasmid pUNO-hDCSIGN2a (InvivoGen Inc., San Diego, CA, USA). The transfection of L-SIGN into SK-HEP-1 cells (BCRC no. 67005) was performed using LipofectamineTM reagents (Invitrogen Inc., Carlsbad, CA, USA) following the manufacturer’s instructions. Stably transfected cells were selected by culturing in 250 μg/mL G418 for 10 days. The presence of L-SIGN mRNA and the surface expression of L-SIGN in these cells was confirmed via RT-PCR and flow cytometry, respectively. Subsequently, these cells were subjected to functional validation to assess the role of the receptor in DENV infection.

### 2.3. Immunodetection of L-SIGN Expression

L-SIGN expression was detected using a flow cytometer with anti-L-SIGN antibodies (FAB162P, R&D Systems, Minneapolis, Minn, USA). Cells were washed twice with phosphate-buffered saline (PBS) and subsequently incubated with antibodies at a 1:50 dilution for 30 min at 4 °C. Following incubation, cells were washed and fixed with paraformaldehyde before fluorescence-activated cell sorting (FACS) analysis, performed using BD Biosciences equipment and data processing with CellQuest software version 3.0 (Becton Dickinson). For RT-PCR analysis, cellular RNA was extracted using Trizol (Invitrogen, Carlsbad, CA, USA) and reverse transcribed into complementary DNA (cDNA) using oligo dT primers (Promega, Madison, WI, USA) and Moloney murine leukemia virus (MMLV) reverse transcriptase (Promega). The PCR was amplified using the forward primer 5′-CAA CAA CCA GTG GCA TCA GA-3′ and the reverse primer 5′-GGC CAT GTA TCT GCT GGA AT-3′.

### 2.4. Measurement of Chemokines and Viral Replication in SK-HEP-1

To assess chemokine production, SK-HEP-1 cells transfected with L-SIGN were seeded in triplicate onto 24-well tissue culture plates and incubated at 37 °C in a humidified atmosphere containing 5% CO₂, either in the presence or absence of DENV-2. After 4 h, cell-free culture supernatants were collected and analyzed for MCP-1 (eBioscience Inc., San Diego, CA, USA) and IP-10 (R&D Systems, Minneapolis, MN, USA) concentrations using enzyme-linked immunosorbent assay (ELISA). The results were calculated from interpolation in a standard curve made from a series of well-known concentrations of standards [27]. The RNA samples extracted from the plasma of patients with DENV-2 infection were stored at −80 °C until further use, as previously described [28]. The viral load was measured by one-step real-time RT-PCR using the ABI 7500 quantitative PCR machine (Applied Biosystems, Perkin-Elmer, Foster City, CA, USA) for 40 cycles using TaqMan technology. The forward primer, reverse primer, and nested fluorescent-probe sequence for detecting DENV-2 are listed as follows: 5′-GGC TTA GCG CTC ACA TCC A-3′; 5′-GCT GGC CAC CCT CTC TTC TT-3′; and 5′-FAM-CGC CCA CCA CTA TAG CTG CCG GA-TAMRA-3′, as previously described [28].

### 2.5. SK-HEP-1 Cells Infection with DEN-2 and Inhibited with MCP-1

Unless otherwise stated, SK-HEP-1 cells with or without L-SIGN transfection were infected with DENV-2 at an MOI of 5 for 2 h at 37 °C and 5% CO_2_. Cells were pre-incubated with DMEM medium alone or with DMEM medium plus recombinant MCP-1 protein (R&D Systems, Minneapolis, MN, USA) at 0.5 or 50 ng/mL for 1 h. After washing twice, cells were infected with DENV-2, washed twice to remove the cell-free virus with PBS, and cultured in complete DMEM medium at a density of 2 × 10^5^ cells/mL in 24-well plates. Cells and supernatants were harvested and analyzed at 24 h post-infection.

### 2.6. Determination of Viral Antigen by Flow Cytometry in SK-HEP-1 Cells

The DENV-2-infected cells were fixed and permeabilized with 2% paraformaldehyde and 0.1% Triton X-100. The permeabilized cells were washed and incubated with mouse MAb (3H5) against DENV-2, used to quantify the infected cells, for 30 min at 4–8 °C. The cells were washed with PBS and incubated with goat F(ab’)2 anti-mouse IgG-FITC (Invitrogen immunodetection, USA) for 30 min at room temperature. Following incubation, the cells were washed twice in PBS and then analyzed using a FACScan flow cytometer (Becton Dickinson FACSCalibur System, USA). The mock-infected SK-HEP-1 cells were run in parallel and served as negative controls. At least 10,000 cells were analyzed using a flow cytometer. Data were analyzed using CellQuest software (Becton Dickinson, San José, CA, USA). The percentage of positive cells and the average fluorescence intensities were determined from FITC fluorescence histograms using a region that was defined based on the analysis of the mock-infected control cells).

### 2.7. Statistical Analyses

For all in vitro studies, statistical significance was determined using Student’s two-tailed *t*-test. A *p*-value of less than 0.05 was considered statistically significant. Data were computed and analyzed using SPSS software (version 13.0).

## 3. Results

### 3.1. Elevated L-SIGN Expression Enhanced DENV-2 Replication than MDDCs with DC-SIGN Expression

Regarding DC-SIGN and L-SIGN, it is essential to recognize their distinct roles and expression patterns in the context of DENV infection. DC-SIGN (dendritic cell-specific intercellular adhesion molecule-3-grabbing non-integrin) serves as a receptor for DENV on the surface of dendritic cells, facilitating viral entry and modulating the subsequent immune response [29]. In this study, we first utilized a stable transfectant of L-SIGN in endothelial SK-HEP-1 cell lines and monocyte-derived dendritic cells (MDDCs) expressing DC-SIGN. Immature MDDCs and SK-HEP-1 cells, with or without L-SIGN cDNA, were incubated for two hours in 24-well plates with supernatants containing DENV at a MOI of 5. The infectivity of parental SK-HEP-1 cells with DENV was relatively lower than that of L-SIGN transfectants but higher than that observed in MDDCs (Figure 1A). To assess DENV-2 replication, we measured the kinetic viral load in SK-HEP-1 cells with or without L-SIGN at 24 and 48 h post-infection. Our findings demonstrate that L-SIGN expression significantly enhances the infectivity of this DENV-susceptible cell line. These data indicate that L-SIGN promotes increased replication of DENV-2 (Figure 1B).

### 3.2. Suppression of Chemokine MCP-1 but Not IP-10 Production in DEN-Infected SK-HEP-1 Cells with L-SIGN

We also measured the concentrations of two chemokines, MCP-1 and IP-10, which have been implicated in the recruitment and activation of monocytes, macrophages, dendritic cells, natural killer (NK) cells, and T lymphocytes [30]. To determine whether higher cell-surface L-SIGN expression correlates with the immune response, we analyzed the kinetic production of these chemokines in SK-HEP-1 cells expressing L-SIGN. It was found that MCP-1, but not IP-10, production was significantly suppressed in SK-HEP-1 with L-SIGN compared to that of parental cells at 48 and 72 h post-infection (1837.86 ± 137.26 pg/mL vs. 463.20 ± 42.14 pg/mL, 2633.98 ± 104.79 pg/mL vs. 1104.93 ± 105.86 pg/mL; *p* = < 0.001 and < 0.001, respectively; Figure 2A). The IP-10 levels showed no significant difference between SK-HEP-1 cells with and without L-SIGN (6.02 ± 1.19 pg/mL vs. 6.25 ± 2.15 pg/mL, 4.84 ± 0.42 pg/mL vs. 6.34 ± 3.30 pg/mL; Figure 2B).

### 3.3. MCP-1 Production by SK-HEP-1 Cells with L-SIGN Involved in Viral Replication of DEN Infection

MCP-1, produced by non-infected bystander dendritic cells (DCs) in response to DENV infection, is a potent chemoattractant for activated T and NK cells and plays a role in modulating the adaptive immune response [30]. Additionally, MCP-3 has been reported to inhibit HIV replication in immortalized cells [31]. In our SK-HEP-1 model, L-SIGN-transfected cells exhibited lower MCP-1 production but higher DENV-2 replication (Figure 2A). Based on these findings, we hypothesized that DENV-infected SK-HEP-1 cells expressing L-SIGN produce lower MCP-1 levels, which may otherwise play a role in restricting viral entry or replication in SK-HEP-1 cells. To test this hypothesis, we utilized recombinant MCP-1 protein to block endogenous DENV-2 replication in SK-HEP-1 cells and observed that viral replication at 48 h post-infection was significantly inhibited in the presence of MCP-1 compared to its absence (Figure 3A,B).

## 4. Discussion

The pathogenesis of DENV infection is a complex interplay of viral, host genetic, and immunological factors. Pattern recognition receptors (PRRs), such as CD209 (DC-SIGN) and its homolog L-SIGN, play a pivotal role in initiating the immune response to pathogens [32,33,34]. While DC-SIGN is predominantly expressed on dendritic cells, L-SIGN is primarily found on liver sinusoidal endothelial cells. Previous studies have implicated L-SIGN in the pathogenesis of other viral infections, suggesting its potential involvement in DENV disease progression [35]. Using PCR genotyping and stable cell lines exhibiting L-SIGN variants, we also demonstrated a higher prevalence of DHF with an increased frequency of L-SIGN neck-9 tandem repeats in the Taiwanese population [26].

Endothelial dysfunction is a hallmark of DHF [36,37]. Our data demonstrate that DENV-2 exhibits a higher replication rate in endothelial cells compared to dendritic cells, supporting the notion that endothelial cells are primary targets for DENV infection and replication. This preferential tropism may contribute to the vascular permeability and organ dysfunction characteristic of severe dengue. Our findings indicate a correlation between L-SIGN expression, DENV replication, and disease severity. Specifically, we observed enhanced DENV replication in liver endothelial cells expressing L-SIGN compared to dendritic cells expressing DC-SIGN. Furthermore, patients with DHF exhibited a higher prevalence of L-SIGN neck-9 tandem repeats [26]. These results suggest that L-SIGN may contribute to the pathogenesis of DHF by facilitating viral replication and potentially modulating the host immune response.

A comprehensive analysis of host inflammatory biomarkers revealed elevated levels of various cytokines in dengue patients, with some correlating with disease severity [38]. Notably, MCP-1 levels were significantly increased in patients with DHF and dengue shock syndrome. Our findings suggest that MCP-1 may play a role in regulating vascular permeability [39] and could be a potential therapeutic target [40].

Interestingly, we observed suppression of MCP-1 production in DENV-2-infected liver endothelial cells expressing L-SIGN, whereas IP-10 levels remained unchanged. This suppression of MCP-1, a potent chemoattractant for monocytes and a key player in the immune response, might be linked to the enhanced viral replication observed in cells with high L-SIGN expression. The present study demonstrated that SK-HEP-1 cells overexpressing L-SIGN promoted DENV-2 replication (as shown in the control portion of Figure 3B), whereas MCP-1 itself had little effect on viral replication. MCP-1 at 0.5 ng/mL maintained high viral replication, whereas MCP-1 at 5 ng/mL reduced infection. This suggests that MCP-1 may regulate L-SIGN stability in a dose-dependent manner or affect downstream viral replication processes. This finding suggests a potential mechanism by which L-SIGN contributes to DENV-2 pathogenesis. The reduced MCP-1 levels may impair the recruitment of immune cells, creating a favorable environment for viral replication and disease progression [41].

Our data support the potential therapeutic utility of recombinant MCP-1 protein in DENV infection. By increasing MCP-1 levels, we observed a reduction in DENV-2 replication and a potential attenuation of DHF-related symptoms. The limitations of the study primarily lie in its focus on the DENV-2 virus, and the findings may not be universally applicable to all dengue virus serotypes. This research could be expanded to include a larger dataset, potentially covering a broader range of conditions or additional variables that were not considered in the current study. Furthermore, future research could validate these findings through animal studies, and these findings warrant further investigation into MCP-1-based therapies for dengue.

## 5. Conclusions

This study demonstrated that dengue hemorrhagic fever (DHF) was associated with a higher viral load and lower MCP-1 levels in dengue virus-infected liver endothelial cells expressing L-SIGN. In vitro studies further revealed that liver endothelial cells infected with dengue virus and exhibiting lower MCP-1 expression were more susceptible to viral replication. Additionally, treatment with exogenous MCP-1 (5 ng/mL) resulted in a significant reduction in viral replication in these cells. Our findings underscore the potential of L-SIGN as a therapeutic target and provide insights into the complex interplay between the virus, host, and immune response in dengue disease. Future research should focus on elucidating the precise mechanisms underlying L-SIGN-mediated DENV pathogenesis and exploring the therapeutic potential of MCP-1 modulation.

## Figures and Tables

**Figure 1 viruses-17-00344-f001:**
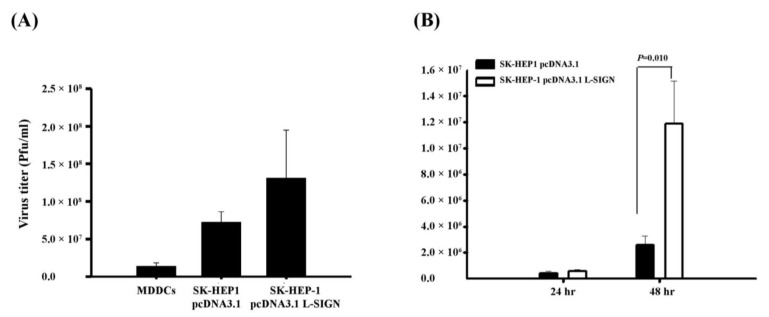
Viral replication from DENV-infected MDDCs and SK-HEP-1 with or without L-SIGN transfection. (**A**) DENV-2 replication (virus copies per 10^5^ cells) in MDDCs and SK-HEP-1 with or without L-SIGN transfection was assessed after infection at MOI of 5 for 48 h. Viral replication was significantly higher in SK-HEP-1 cells with L-SIGN transfection than in those without L-SIGN. (**B**) Kinetic changes of DENV-2 replication (virus copies per 10^5^ cells) were assessed after infection at MOI of 5 for 24 to 48 h. Viral replication was significantly higher in SK-HEP-1 cells from transfectants with L-SIGN than in parental cells at 24 h and 48 h post-infection.

**Figure 2 viruses-17-00344-f002:**
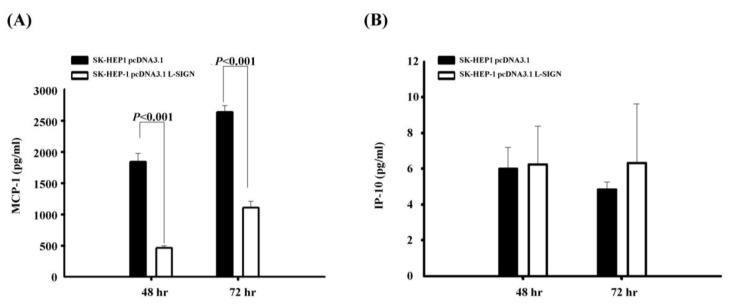
Kinetic chemokine production by DENV-infected SK-HEP-1 cells with or without L-SIGN transfection. The supernatants of MCP-1 and IP-10 levels were measured using ELISA after DENV infection at MOI of 5 for 48 to 72 h. The production of MCP-1 (**A**) levels after DENV-2 infection was significantly suppressed in SK-HEP-1 cells with L-SIGN transfection compared to those without L-SIGN. The production of IP-10 (**B**) after DENV-2 infection was not significantly different between the two groups. Data are presented as mean ± SE calculated from five pairs of samples.

**Figure 3 viruses-17-00344-f003:**
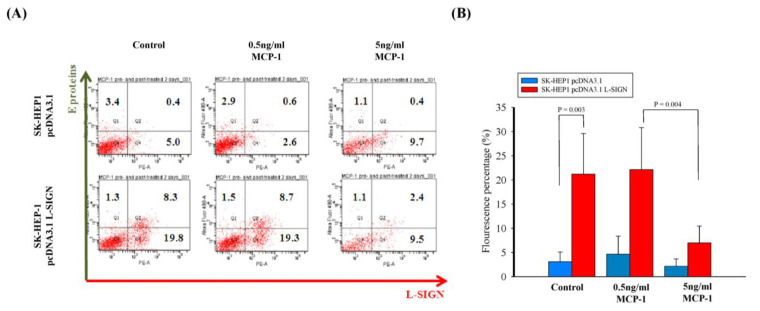
Recombinant MCP-1 protein in DENV replication. (**A**) Flow cytometry was used to detect the dengue virus envelope protein (E) and L-SIGN expression on the cell surface in DENV-infected SK-HEP-1 pcDNA3.1 and SK-HEP-1 pcDNA3.1 L-SIGN with 0, 0.5, and 5 ng/mL MCP-1. (**B**) DENV copy numbers were counted by fluorescence intensity in DENV-infected SK-HEP-1 pcDNA3.1 and SK-HEP-1 pcDNA3.1 L-SIGN with 0, 0.5, and 5 ng/mL MCP-1.

## Data Availability

The data and materials presented in this investigation are available by request from the corresponding author.

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
