# Peer review of "MCP-1 Reduction by L-SIGN Expression in Dengue Virus-Infected Liver Endothelial Cells"

_viruses, 2025, doi:10.3390/v17030344_

Round 1
Reviewer 1 Report
Comments and Suggestions for Authors
This is an interesting work whose findings Indicate a correlation between L-SIGN expression, Dengue virus (DV) replication and disease severity; expanding the knowledge of DV infection and with implications for new therapeutic strategies development.
However, some questions regarding this article still remain and need to be clarified:
1. Intracellular cytometry for indirect detection of viral antigens can be a technique that requires a lot of controls and care, and can often present a relatively low number for the percentage of positive cells. Given this scenario, could the authors explain why they chose to evaluate only 10,000 events, when an analysis of at least 50,000 events per tube would probably be the most appropriate in this case?
2. Discuss whether data obtained for DV2 may apply to other serotypes with the same intensity? The authors have some new experimental data about it ?
3. In figure 2B, the levels of CXCL-10 (IP-10) produced seem very low in relation to other experimental models present in the literature. Could the authors comment on this ? IFNg is an extremely important molecule in the response to DV and the main inducer of IP-10 (inducible protein 10). Could the authors comment on why they did not evaluate the same parameters in the presence of IFNg ?
4. Figure 2 shows that "..... 10 subjetcs....". Wasn't it a cell line? Or would it be an experimental "n" of ten? The authors should clarify this point.
5. Figure 3A (cytometry) is a representative figure of how many experiments?
6. In Figure 3B, the statistical differences are not indicated.
7. MCP-1 may be affecting L-SIGN expression in SK-HEP-1 pcDNA3.1 L-SIGN cells and more strongly than in SK-HEP1 pcDNA3.1 cells (figure 3, MCP-1 - 0.5 ng/ml). The authors may discuss this phenomenon.
8. In the final part of the discussion, the authors could mention the limitations of the study and its perspectives
Author Response
- Intracellular cytometry for indirect detection of viral antigens can be a technique that requires a lot of controls and care, and can often present a relatively low number for the percentage of positive cells. Given this scenario, could the authors explain why they chose to evaluate only 10,000 events, when an analysis of at least 50,000 events per tube would probably be the most appropriate in this case?
Ans: Thank you for your thorough review. As you suggested, 50,000 events would indeed be more representative than 10,000 events. Sometimes, due to limited specimen availability, we are compelled to analyze fewer events to represent our research findings. Fortunately, in this experiment, we determined that 10,000 events were sufficient to demonstrate the differences observed in SK-HEP-1 cells.
- Discuss whether data obtained for DV2 may apply to other serotypes with the same intensity? The authors have some new experimental data about it ?
Ans: Thank you for your expert input. To our knowledge, DENV-2 is the serotype most frequently associated with severe dengue hemorrhagic fever among the four dengue virus serotypes. To avoid misinterpretation and given that we cannot reinitiate experiments with other serotypes, we have revised the term "DENV" to "DENV-2" on page 7 of the discussion section (highlighted in red).
- In figure 2B, the levels of CXCL-10 (IP-10) produced seem very low in relation to other experimental models present in the literature. Could the authors comment on this ? IFNg is an extremely important molecule in the response to DV and the main inducer of IP-10 (inducible protein 10). Could the authors comment on why they did not evaluate the same parameters in the presence of IFNg ?
Ans: Thank you for your insightful query. In our previously published research, we demonstrated that the L-SIGN neck-region 9-tandem repeat allele was associated with lower IFN-γ levels. Therefore, in this follow-up study, we chose to focus on IP-10, which is associated with hepatic adenocarcinoma cells, rather than IFN-γ.
- Figure 2 shows that "..... 10 subjetcs....". Wasn't it a cell line? Or would it be an experimental "n" of ten? The authors should clarify this point.
Ans: Thank you for your meticulous review. Indeed, this was our typographical error. We have corrected it to "Data are presented as mean ± SE calculated from five pairs of samples" (highlighted in red).
- Figure 3A (cytometry) is a representative figure of how many experiments?
- In Figure 3B, the statistical differences are not indicated.
Ans: Thank you for your pertinent query. Following your suggestion, we have reviewed the results from other replicate experiments. This experiment was performed with five replicates, and accordingly, we have updated the figure to include statistical P values calculated using Student's t-test for comparisons between two key groups.
- MCP-1 may be affecting L-SIGN expression in SK-HEP-1 pcDNA3.1 L-SIGN cells and more strongly than in SK-HEP1 pcDNA3.1 cells (figure 3, MCP-1 - 0.5 ng/ml). The authors may discuss this phenomenon.
Ans: Thank you for your insightful query and for noting the effect of MCP-1 on DENV-2 replication. This study demonstrated that SK-HEP-1 cells with overexpression of L-SIGN promote DENV-2 replication (as shown in the control portion of Figure 3B), however, MCP-1 has little effect on viral replication. While MCP-1 at 0.5 ng/ml maintains high viral replication, 5 ng/ml reduces infection. This suggests that MCP-1 may regulate L-SIGN stability or impact downstream viral replication processes in a dose-dependent manner. We have added the above explanation to the discussion section, highlighted in red (page 7, lines 27-32).
- In the final part of the discussion, the authors could mention the limitations of the study and its perspectives.
Ans: Thank you for your thorough review. The limitations of the study primarily lie in its focus on DENV-2 virus, and the findings may not be universally applicable to all dengue virus serotypes. Regarding Perspectives and Future Directions, this research could be expanded to include a larger dataset, potentially covering a broader range of conditions or additional variables that were not considered in the current study. Furthermore, future research could validate these findings through animal studies and potentially extend them to clinical applications. We have added these points to the discussion section, highlighted in red (page 7, lines 38-43).
Reviewer 2 Report
Comments and Suggestions for Authors
Dear Authors,
After reviewing the manuscript, I identified specific areas requiring corrections and explanations.
The methodology needs to be more detailed.
Please refer to the comments concerning cell culturing:
· Please provide the manufacturer of RPMI
· Was FBS used for MDDC culturing? If so, please provide details. Was the RPMI used for MDDC supplemented with antibiotics?
· From methodology – “medium supplemented with cytokines” Please explain what cytokines were used. Where did they originate from?
· Provide the origin of U937, K562 and SK-HEP-1 cells
· You’ve mentioned using U937 cells. What was the purpose of using these cells in this research? K562 cells were used further in experiments (2.6. Determination of viral antigen by flow cytometry in K562 cells), but there is no information on using U937 in any experiments.
In 2.3. Immunodetection of L-SIGN expression
a. Please provide details on the reverse transcription, kit manufacturer
b. Please correct “The PCR was amplified by PCR using the forward primer 5′- CAA CAA CCA GTG GCA TCA GA-3′ and the reverse primer 5′- GGC CAT GTA TCT GCT GGA AT-3′ ”.
In 2.4. Measurement of chemokines and viral replication in SK-HEP-1, please provide details on the real-time RT-PCR analysis kit. What about reverse transcription? Was it a one or two-step reaction?
At the end of the provided manuscript, you have responded “not applicable” to “Institutional Review Board Statement” and “Informed Consent Statement”. Unfortunately, I cannot agree with this statement.
This study involved using human blood to isolate peripheral blood mononuclear cells. That is why the authors must declare that the investigations were carried out following the rules of the Declaration of Helsinki of 1975 (revised in 2013). According to point 23 of this declaration, you should provide the Institutional Review Board Statement or confirmation from another appropriate ethics committee.
Regarding blood donors, a statement of informed consent for participation in research should also be acquired.
Please write species, like Mycobacterium tuberculosis, in italics.
Superscripts or subscripts need to be applied in numerous places throughout the text.
Author Response
Thank you for your detailed review. We have made the following corrections and additions:
- We have added the manufacturer of RPMI and confirmed that FBS and antibiotics were included in the MDDC cell culture medium, highlighted in red on page 3, lines 9-10.
- We apologize for any confusion regarding the cytokines. As this was an extension of the previous sentence referring to GM-CSF and IL-4, we have revised "cytokines" to "rhGM-CSF and rhIL-4" to clarify their human origin.
- We acknowledge that U937 and K562 cells were not used in this study. This was our oversight, and these references have been struck through. SK-HEP-1 cells were obtained from BCRC, as now highlighted in red on page 3, lines 27-28.
- Following from the above, we have corrected the section heading from "2.6. Determination of viral antigen by flow cytometry in K562 cells" to "2.6. Determination of viral antigen by flow cytometry in SK-HEP-1 cells."
- We have added that oligo dT and MMLV reverse transcriptase were from Promega, highlighted in red on page 3, lines 40-41.
- We have removed the redundant "by PCR" in the primer description sentence, as indicated by strikethrough.
- Regarding dengue virus RT-PCR, while we published this methodology in 2001 (Chen RF, et al. FEMS Immunol Med Microbiol. 2001 Feb;30(1):1-7), we have added "one-step" before real-time RT-PCR (in red) and included "as previously described (41)" after the primer and probe sequences to avoid any misunderstanding.
- Following your reminder about IRB and Informed Consent Statement, we have added the relevant information in the final paragraph, highlighted in red.
- We have italicized Mycobacterium tuberculosis (page 2, line 32) and corrected all superscripts and subscripts throughout the manuscript.
Thank you again for your patience and thorough guidance.
Reviewer 3 Report
Comments and Suggestions for Authors
Thank you for sharing your article on the MCP-1 reduction by L-SIGN expression in endothelial cells infected with dengue virus. The following minor comments may help when revising the manuscript.
General comment
-Please make sure that the meaning of all abbreviations used throughout your manuscript is explained.
-Please check the journal's requirements whether it suffices to state the manufacturers of laboratory items used or if cat. no. have to be provided too.
-Please enlarge the font size throughout your figures.
-Consider separating discussion and conclusion sections.
Section 2.1
-Please provide more details on the FACS analysis performed, including device specifications.
Section 2.4
-More detail is needed on the TaqMan technology used beside the stated primers.
Section 2.5
-Please be more specific which medium you used for pre-incubation and which solution was used for washing the cells. The same applies in part to section 2.6.
Author Response
- Please make sure that the meaning of all abbreviations used throughout your manuscript is explained.
Ans: Thank you for your attention to detail. We have added recombinant human granulocyte macrophage-colony stimulating factor (rhGM-CSF) and recombinant human interleukin-4 (rhIL-4), which are highlighted in red on page 3, lines 11-12.
- Please check the journal's requirements whether it suffices to state the manufacturers of laboratory items used or if cat. no. have to be provided too.
Ans: Thank you for your suggestion. We have supplemented the information regarding the manufacturers of materials, highlighted in red. Additionally, we have re-evaluated the journal's requirements for the Materials and Methods section, which states, "They should be described with sufficient detail to allow others to replicate and build on published results. New methods and protocols should be described in detail while well-established methods can be briefly described and appropriately cited. Give the name and version of any software used and make clear whether computer code used is available. Include any pre-registration codes." Therefore, please understand that we cannot provide the catalog numbers for the products.
- Please enlarge the font size throughout your figures.
Ans: Thank you for your suggestion. We have enlarged all figures, thus increasing the font size accordingly.
- Consider separating discussion and conclusion sections.
Ans: Thank you for your suggestion. We have now separated the conclusion section from the discussion section.
Section 2.1
- Please provide more details on the FACS analysis performed, including device specifications.
Ans: Thank you for your attention to detail. We have added "Enriched CD14+ cells (purity > 95% as determined by FACS analysis using BD Biosciences and data processing with CellQuest software from Becton Dickinson)" highlighted in red on page 3, lines 7-8.
Section 2.4
- More detail is needed on the TaqMan technology used beside the stated primers.
Ans: Thank you for your kind suggestion. TaqMan technology has been in use for over 20 years; we published this methodology in 2001 (Chen RF, et al. FEMS Immunol Med Microbiol. 2001 Feb;30(1):1-7). Therefore, please forgive us for providing only the reference following the primers without further describing the TaqMan technology to conserve space.
Section 2.5
- Please be more specific which medium you used for pre-incubation and which solution was used for washing the cells. The same applies in part to section 2.6.
Ans: Thank you for your attention to detail. We have added the names of the corresponding medium and the washing solution, which are highlighted in red.
Round 2
Reviewer 2 Report
Comments and Suggestions for Authors
Dear Authors, thank you for responding to my previous comments.
Author Response
Thank you for the positive feedback from your comments.